# Combining Coordination and Hydrogen Bonds to Develop Discrete Supramolecular Metalla-Assemblies

**Bruno Therrien**

Institute of Chemistry, University of Neuchatel, Avenue de Bellevaux 51, CH-2000 Neuchatel, Switzerland; bruno.therrien@unine.ch; Tel.: +41-32-718-2499

**Abstract:** In Nature, metal ions play critical roles at different levels, and they are often found in proteins. Therefore, metal ions are naturally incorporated in hydrogen-bonded systems. In addition, the combination of metal coordination and hydrogen bonds have been used extensively to develop supramolecular materials. However, despite this win-win combination between coordination and hydrogen bonds in many supramolecular systems, the same combination remains scarce in the field of coordination-driven self-assemblies. Indeed, as illustrated in this mini-review, only a few discrete supramolecular metalla-assemblies combining coordination and hydrogen bonds can be found in the literature, but that figure might change rapidly.

**Keywords:** coordination chemistry; hydrogen bonds; supramolecular chemistry; metalla-assemblies; coordination-driven self-assembly; orthogonality; ligands; metal ions

## 1. Introduction

The preparation and characterization of discrete metal-based assemblies have been the focus of several research groups. Such metalla-assemblies are obtained by combining metal ions and multidentate ligands in a pre-designed and controlled manner [1–5]. These supramolecular metalla-assemblies can be used as sensors [6–8], anticancer agents [9,10], hosts for guest molecules [11,12], drug carriers [13], mesogens [14,15], or molecular flasks [16,17]. About 40 years ago, the first coordination-driven metal-based squares (Figure 1), composed of linear diphosphine ligands and tetracarbonyl metal ions (Cr, Mo, W), were synthesized [18].

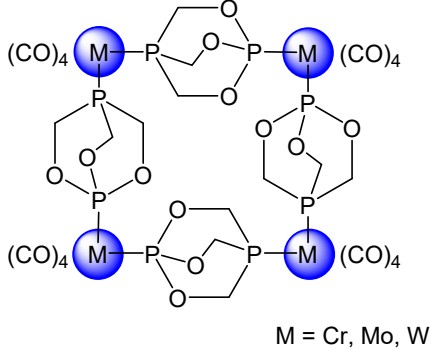

M = Cr, Mo, W

**Figure 1.** Molecular structure of the first coordination-linked metalla-squares [18].

A few years later, the field really took off with the introduction of 90° square-planar palladium ions, which are versatile building blocks in supramolecular chemistry [19]. Nowadays, all kind of transition

metal ions with different coordination geometries have been introduced in metalla-assemblies, and the field is flourishing.

Like the field of coordination-driven self-assemblies, hydrogen-bonded self-assembled systems have received for many years a great deal of attention [20–25]. The directionality, stability, reversibility, and biological importance of hydrogen bonds have encouraged research groups to use hydrogen-bonded motifs to construct supramolecular assemblies. An appropriate selection of donor and acceptor groups, in a pre-organized fashion, can control the strength and geometry of the designed supramolecular structures, and accordingly, allow the formation of two-dimensional and three-dimensional assemblies. Nowadays, the degree of sophistication has reached an incredible level, far beyond the simple DNA helices, and beautiful examples are published at a regular pace with different applications in mind [20–25].

In the field of materials science, coordination and hydrogen bonds have been joint to generate polymers, dendrimers, and other supramolecular assemblies [26–32]. An early example of such materials comes from the group of Reinhoudt, in which a barbituric acid entity was coupled to palladium-based metallo-dendrons to generate metallo-dendrimers [33]. In these systems, the barbituric acid residue forms a rosette type structure via hydrogen bonds, while the supramolecular network is further extended by the dendritic arms: The two functions are linked together by coordination chemistry. Following this pioneer report, similar combinations have been used to develop coordination and hydrogen-bonded materials [26–35].

Surprisingly, despite this relative popularity, the combination of coordination and hydrogen bonds to form discrete supramolecular metalla-assemblies remains scarce. Most examples are limited to cyclic and planar entities (one and two dimensions), and only recently, systems showing cavities and cage-like structures (three dimensions) have appeared in the literature. These hybrid self-assembled systems involving coordination chemistry and hydrogen-bonded interactions to form discrete entities are presented and discussed in this short review, thus showing the great potential of combining coordination and hydrogen bonds to develop new supramolecular metalla-assemblies.

## 2. Planar and Macrocyclic Assemblies Exploiting Coordination and Hydrogen Bonds

In crystal engineering, the combination of metal ions and hydrogen bonds has been extensively explored [36], and the first examples of discrete coordination and hydrogen-bonded systems were probably inspired by solid-state chemistry. Joining several metal-based chromophores is needed for the preparation of light-harvesting systems, however, to better understand the electronic pathway and metal-metal communications involved in such systems, having a dinuclear compound can be more appropriate. With that in mind, the groups of Ward and Barigelletti have studied the electronic energy transfer process between metal-polypyridyl complexes linked by complementary hydrogen-bonded groups [37,38]. Bispyridyl ligands functionalized with nucleobases were synthesized and used to connect two metal ions, see Figure 2.

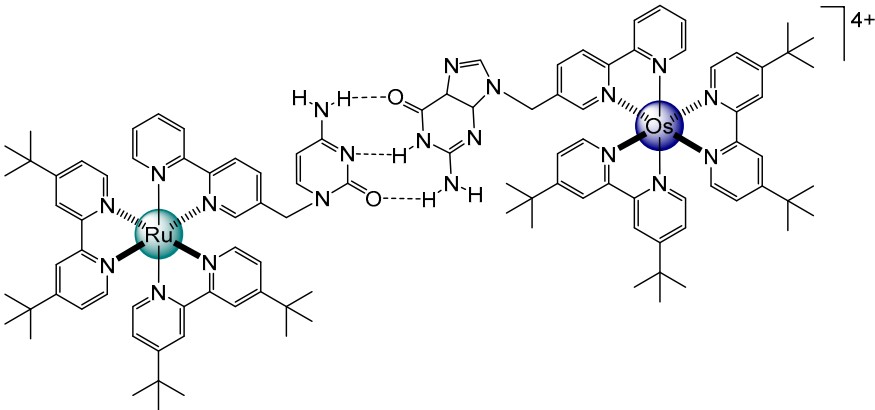

**Figure 2.** Cationic hydrogen-bonded dinuclear systems showing metal-metal energy transfer [37].

The relatively high binding constant of the nucleotide base pairs and the nature of the dinuclear systems have suggested that the energy transfer occurs, even in solution (CH$_2$Cl$_2$), via the hydrogen-bonded interface.

A similar bis-rhodium complex has been synthesized [39], and a single crystal X-ray structure analysis has confirmed the dimeric nature of the system (Figure 3). Interestingly, upon coordination to the rhodium pentamethylcyclopentadienyl unit, the hydrogen bond pairing between two 7-diphenylphosphino-1*H*-quinolin-2-one ligands is not disturbed. Diffusion-ordered NMR spectroscopy in CD$_2$Cl$_2$ shows that the dimeric structure is stable in aprotic solvents.

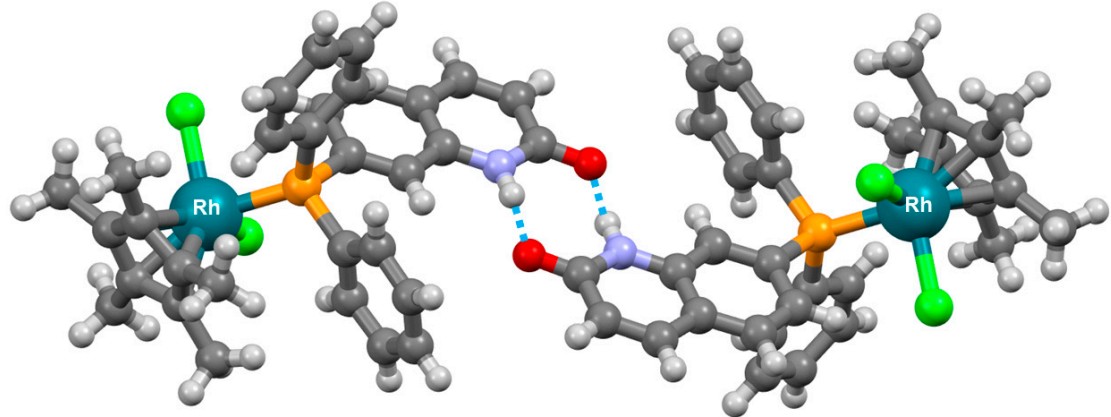

**Figure 3.** Dimeric structure of a hydrogen-bonded bis-rhodium complex [39].

Other dinuclear systems have been prepared, in which a combination of coordination and hydrogen-bonded interactions were used. For instance, a platinum-based dimer has been prepared in view to synthesize higher nuclearity systems [40]. Unfortunately, the self-complementary quinolone hydrogen bonds were too weak compared to the π-stacking interactions of the ligands, thus forming in solution a coordination macrocycle instead of a hydrogen-bonded tetranuclear system (Scheme 1).

**Scheme 1.** Self-assembly of a coordination macrocycle (**top**) over a tetranuclear hydrogen-bonded system (**bottom**) [40].

Quinolone-based ligands were also used with octahedral metal center. Indeed, a dinuclear rhodium-based complex was obtained by reacting [Cp*RhCl$_2$]$_2$ (Cp* = pentamethylcyclopentadienyl) with 7-diphenylphosphino-1*H*-quinolin-2-one in a 1:2 stoichiometry [39]. The cationic dinuclear

complex (Figure 4) is stable in solution (CD$_2$Cl$_2$), and NMR studies suggest that no dynamic behavior (assembly-disassembly) is occurring at room temperature in aprotic solvent. As emphasized in Figure 4, strong π-π stacking interactions take place, which increases the stability of the macrocyclic structure. Analogous dinuclear systems were obtained by reacting [(*para*-cymene)RuCl$_2$]$_2$ with 1-(4-oxo-6-undecyl-1,4-dihydropyrimidin-2-yl)-3-(pyridine-4-ylethyl)urea (UPy-L) in a 1:2 fashion [41]. The neutral complex (Figure 5) is stable under ESI-MS (electro-spray ionization–mass spectrometry) conditions. The dinuclear complexes were also incorporated in tetranuclear systems, in which the UPy-L units were parallel to each other to generate metalla-rectangles [42].

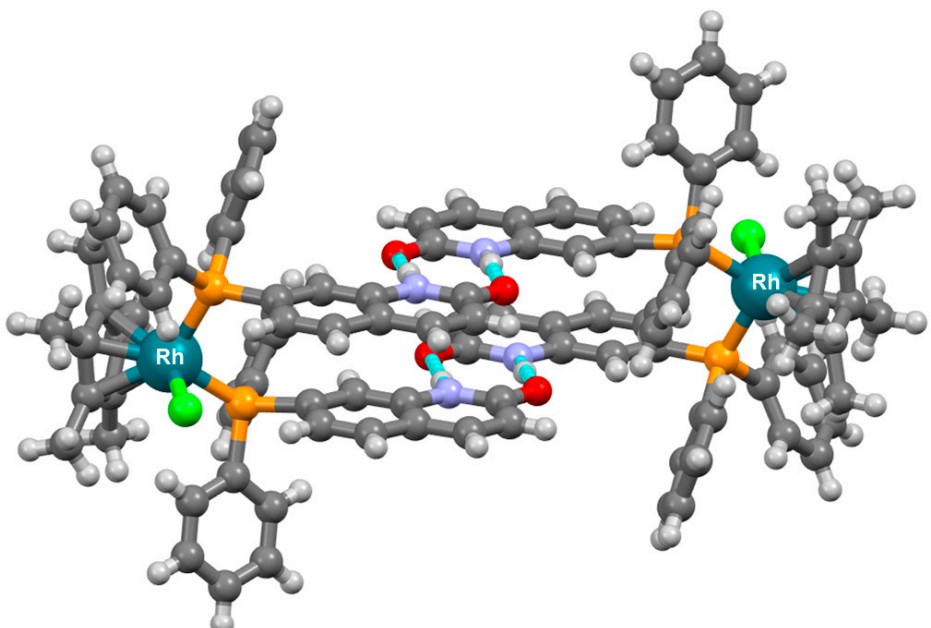

**Figure 4.** Dinuclear rhodium-based metallacycle [39].

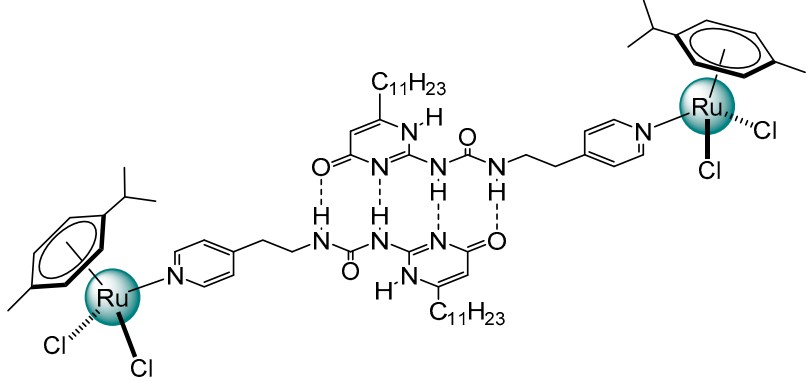

**Figure 5.** A dinuclear complex incorporating piano-stool complexes and derived-ligands allowing hydrogen-bonded assemblies [41].

Two-dimensional assemblies with more than two metal ions have also been synthesized, using for example square-planar complexes. Tetranuclear and hexanuclear platinum-based metalla-cycles were prepared by Rendina and his coworkers [43]. The nicotinic acid pair acts as a 120° bridging ligand, and upon coordination to *trans*-bis(diphenylphosphine)platinum units, it forms a dinuclear sub-unit that can be coupled to other bidentate ligands. In combination with a 180° bidentate spacer (4,4′-biphenyl), a hexanuclear metalla-cycle is obtained (Figure 6A), while in combination with a 120° bidentate ligand (4,4′-benzophenone), a tetranuclear metalla-cycle is isolated (Figure 6B).

When *iso*-nicotinic acid is used instead, oligomeric and polymeric species are formed, demonstrating the importance of ligands and metal ions geometry for the preparation of discrete metalla-assemblies.

**Figure 6.** Hexanuclear (**A**) and tetranuclear (**B**) metalla-cycles built from nicotinic acid and square-planar platinum ions [43].

Hydrogen-bonded dimers of *para*-pyridyl-substituted 2-ureido-4-1*H*-pyrimidinone and *cis*-coordinated palladium complexes have been combined to afford a tetranuclear metalla-cycle [44]. In solution (CDCl₃), a mixture of a metalla-square (Figure 7) and a metalla-triangle was observed. At low concentrations (1 mM), the triangular assembly is favored, while at higher concentrations, the amount of the square-like structure is increasing significantly. This study confirms that the nature of metalla-cycles can be controlled by steric factors, by the solubility of the final entity, and by the geometry of the different building blocks.

**Figure 7.** Molecular structure of a palladium-based tetranuclear assembly [44].

The same *para*-pyridyl-substituted 2-ureido-4-1*H*-pyrimidinone hydrogen-bonded dimer was used to construct tetranuclear arene ruthenium metalla-rectangles [42]. NMR spectroscopy and

DFT calculations showed that the formation of the hydrogen-bonded assembly results in an energy gain of $\Delta E = -146.8$ kJ mol$^{-1}$, thus confirming the stability in solution of these multiple hydrogen-bonded assemblies.

The melamine–cyanuric acid (barbituric acid) pairing is among the most studied hydrogen-bonded system [45–48]. Rosette-type and tapelike structures can be achieved by the controlled functionalization of the sub-units [49,50]. Steric groups will favored the formation of discrete rosette-type structures, while small and highly soluble groups will increase tapelike structures. Therefore, in view to obtain discrete metal-coordinated rosette-type systems, a series of pyridyl-functionalized cyanuric acid [51] and melamine [52] derivatives were synthesized. Coordination of arene ruthenium complexes to the pyridyl groups has generated trinuclear (Figure 8A) and hexanuclear (Figure 8B) rosette-type assemblies.

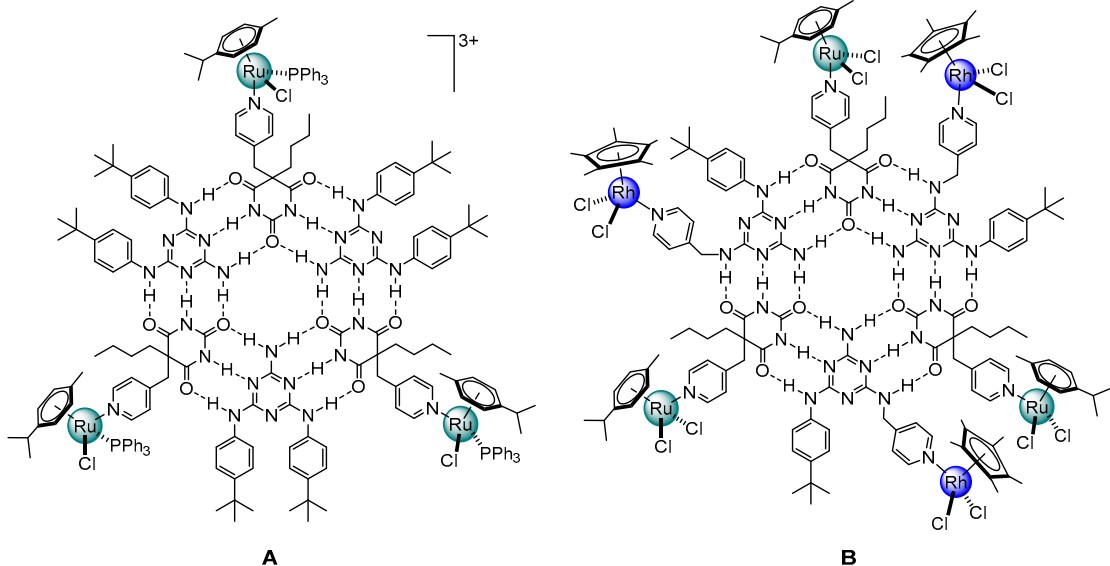

**Figure 8.** Examples of trinuclear (**A**) and hexanuclear (**B**) rosette-type metalla-assemblies [51,52].

The use of metalated nucleobases has been another approach in supramolecular chemistry that combines coordination chemistry and hydrogen bonds [53–59]. In such systems, the natural pairing of nucleic acids is replaced by metal-mediated base pairs, which allows the generation of hybrid DNA structures. Various applications have been foreseen for these derivatives (sensing, expending the genetic code, forming nanoclusters or nanowires, DNA technology) and they have been the subject of many reviews [53–59]. Therefore, this abundant literature will not be covered here, and the readers who are interested in that particular area are encouraged to refer to these reviews to complete the discussion.

## 3. Cage-Like Assemblies Exploiting Coordination and Hydrogen Bonds

Several supramolecular capsules built by a combination of two functionalized $C_2$ symmetrical calix[4]arene cavitands have been synthesized by Yamanaka and his coworkers [60–62]. In these systems, the two capsules are linked by two metal ions and two pairs of hydrogen bonds (Figure 9). The size of the calix[4]arene and the length of the functional groups (hydrogen-bonded derivatives and pyridyl groups) dictate the size of the cavity. In some cases, the cavity is filled by an anion, while in other systems, a guest molecule is trapped. The guest exchange dynamics are linked to the nature of the anions used, and their ability to generate conformational changes by disrupting the intramolecular hydrogen-bonded system.

**Figure 9.** Hybrid capsules built from two functionalized calix[4]arene cavitands [62].

The conical-shape of calix[4]arene was also used to generate giant uranyl-based cages [63]. In these systems, hydrogen bond interactions are exploited to ensure that the calixarene carboxylate ligands adopt a stable and symmetrical conformation, which ultimately forces the carboxylate anion to coordinate to the uranyl cation ($UO_2^{2+}$) in a controlled manner. This strategy has allowed the formation of several discrete icosahedral cage-like structure (Figure 10), in which the metals are not located at the corners or edges of the assemblies, and for which the cavity of the large anionic capsule is relatively well shielded.

The cooperative action of coordination bonds and quadruple hydrogen-bonded interactions has allowed the synthesis of tetrahedron cage-like structures [64]. The symmetry, size, and nature of the assembly are linked to the flexibility of the ligands, the choice of the metal ions ($Hg^{2+}$, $Fe^{2+}$, $Zn^{2+}$) and the conditions used (solvent polarity, concentration, anion, temperature). In the iron(II) derivatives, the quadruple hydrogen-bonded units are linked to 2,2′-bipyridyl group, to produce a tetrahedron cage-like structure (Figure 11). Stability studies have showed that protic solvents (DMSO, $H_2O$) initiate the disassembly of the cage-like structure. However, some derivatives show remarkable stability in

polar solvents, and even in the presence of coordinating competing agents, thus suggesting that these capsules can be used as reactors for catalytic reactions.

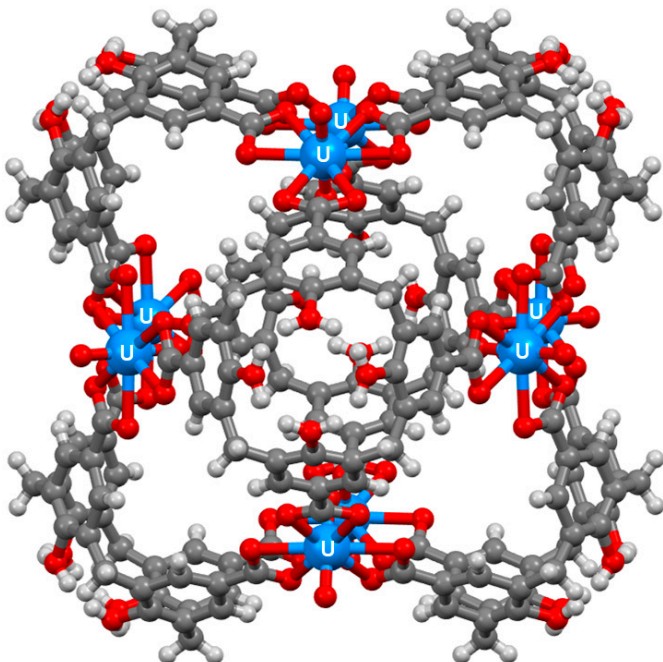

**Figure 10.** Molecular structure of a hexameric cage-like structure built from six calix[4]arene carboxylates and eight uranyl cations [63].

**Figure 11.** Molecular structure of a hydrogen-bonded tetrahedron cage-like system [64].

The nature of the metal ion was also a critical point when dealing with these quadruple hydrogen-bonded units linked to 2,2'-bipyridyl ligand [64]. Replacing $Zn^{2+}$ or $Fe^{2+}$ with $Hg^{2+}$ not only modified the stability, but also the overall geometry. The large ionic radius of $Hg^{2+}$ provides a wider separation of the coordinated 2,2'-bipyridyl ligands, thus allowing the quadruple hydrogen-bonded units to stack on top of each other and to form a triple decker system (Figure 12). The helicate structure is less stable than the tetrahedron systems, as the coordination energy of 2,2'-bipyridyl to $Hg^{2+}$ remains relatively weak compared to $Fe^{2+}$.

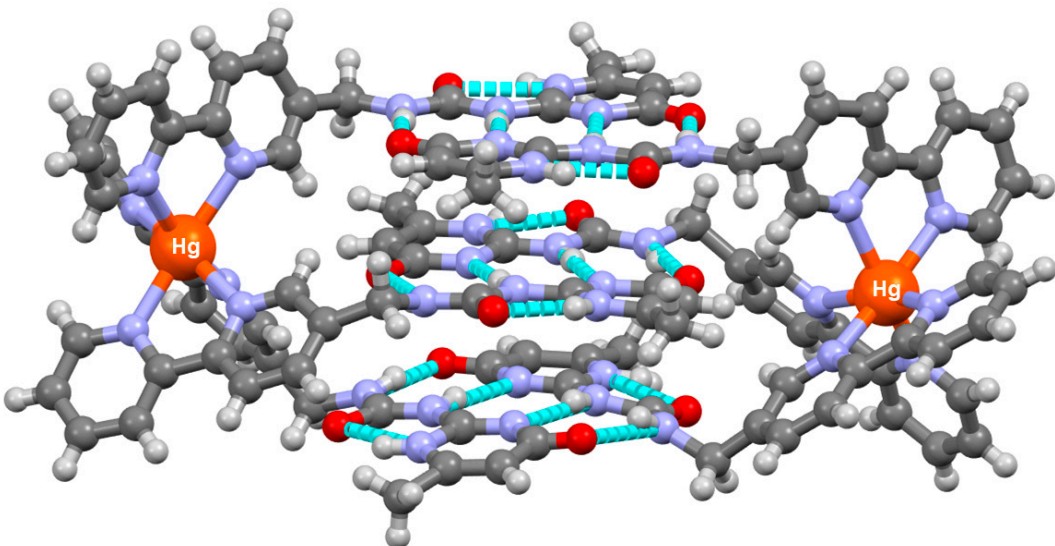

**Figure 12.** Dinuclear mercury-based hydrogen-bonded helicate [64].

## 4. Conclusions

As pointed out here, as well as in several reviews and publications [26–35], allowing two or more non-covalent interactions to take place simultaneously can be quite challenging (orthogonality concept) [30]. To be successful, a compatibility between the hydrogen-bonded and coordination interactions is essential, as individually they show different solubility, different flexibility and different stability. Moreover, to form a discrete supramolecular metalla-assembly, the ligands and the hydrogen bonded units should not compete for the metal ions, and they should cooperate. Therefore, it is not so surprising that so far the number of discrete metalla-assemblies combining coordination and hydrogen bonds remains limited. Nevertheless, we can assume that considering recent progress in the understanding on how such orthogonal concepts can be applied to supramolecular systems, and how innovative strategies have recently emerged in the field, that soon, we will see more of these discrete supramolecular metalla-assemblies.

**Funding:** This research received no external funding.

**Acknowledgments:** The author would like to thank past and present members of his group.

**Conflicts of Interest:** The authors declare no conflict of interest.

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
