# Peer review of "Combining Coordination and Hydrogen Bonds to Develop Discrete Supramolecular Metalla-Assemblies"

_chemistry, doi:10.3390/chemistry2020034_

Round 1
Reviewer 1 Report
The mini-review by Therrien reviews discrete supramolecular assemblies formed through a combination of coordination driven self-assembly and hydrogen bonding. Both of these are pillars of supramolecular chemistry but have been rarely combined in discrete assemblies. Despite still being in its infancy the topic of the review is topical and of wide interest within supramolecular chemistry. The author is well placed to write this review having published widely in the field of metallo-supramolecular assemblies and their biological applications. I recommend publication in Chemistry pending the following minor revisions.
- I would tone down the emphasis on biological systems in the abstract and include more emphasis on why the reviewed assemblies are interesting in their own right since many are not biologically relevant.
- The combination of H-bonding and metal coordination has been used more extensively in the field of supramolecular polymers (eg 1021/jacs.9b02677, 10.1073/pnas.1307472110). This should be commented on in the introduction.
- Considering how few examples there are of discrete assemblies incorporating both types of interaction the authors should comment more on some of the challenges underpinning their assembly.
- While the Chemdraw images of the assemblies are quite clear, it could be nice to add illustrations of the crystal structures to the figures (where available) to better illustrate their 3D arrangement.
Author Response
I would like to thank the Reviewer for the positive comments, and the following changes have been made to improve the quality of the manuscript.
1) The abstract has been modified, and instead on mentioning metallo-enzymes, materials are presented. This is also emphasized in the introduction, by the addition of a new paragraph (See Reviewer 3).
2) These references, and this aspect is now discussed in the introduction (new paragraph, 11 new references.
3) The orthogonally concept (challenge) is now discussed in the revised version of the manuscript.
4) Four new figures, from X-ray structure analysis, have been added.
Reviewer 2 Report
This mini-review by Bruno focuses on an emerging direction in the supramolecular assembly field: discrete assemblies combining both metal-coordination and hydrogen bonding interactions. The research backgrounds, main progresses, important examples along this unique direction have been summarized. The whole manuscript is well-organized and very pleasing to read. This reviewer recommends the publication of fantastic work without any hesitation.Author Response
I would like to thank the Reviewer for the positive comments.
Reviewer 3 Report
Over the last 2-3 decades, self-assembly has emerged as a key topic in supramolecular chemistry. Therefore, many reviews have been published on the use of metal coordination (e.g. ref. [1-5]) or hydrogen bonding ([20-24]) as “gluing” tools, among others, such as hydrophobic or electrostatic interactions. But, as the author cleverly points out, papers highlighting the simultaneous use of metals and hydrogen bods are still scarce. I should add, scarce but rapidly growing. That’s why I consider this review original and timely. Moreover, it is written in a clear and concise way, and the figures are well selected and carefully drawn. Therefore, I strongly recommend publication, subject to minor though relevant modifications. Some of them are necessary to give due credit to some pioneering contributions that I feel missing. Others are aspects that I consider poorly developed regarding the contents of some papers mentioned, because they seem to focus more on what was done instead of why it was done. Finally, as for any topic related to orthogonality, I suggest to introduce a few comments on compatibility of hydrogen bonds with metals (solvents, metals competing for carbonyl groups or similar H-acceptors, or pros and cons regarding flexibility vs. rigidity in both approaches).
In summary, I recommend the author to address the following points before publishing:
1. Missing references: Although an exhaustive survey of contributions is admittedly impossible, I believe that the 1996 Stoddart paper on self-assembly in natural and unnatural systems (ACE 35, 1154), or a couple of Lehn’s articles (not only ref. [51] in the conclusions), reviews or books on the field published during 90’s decade, are a must early in the manuscript.
Even more relevant, since it is directly related to the main topic of this review is the article by Reinhoudt and co-workers in 1997 (ACE, 36, 1006), where it is stated: …“From a synthetic point of view it is important that the two types of interactions [metal–ligand coordination and hydrogen bonding] are “orthogonal”, that is, mutually compatible.”… (mentioned in Zimmerman’s seminal review on orthogonality: Chem. Commun. 2013, 49, 1679), Neither Reinhoudt’s nor Zimmerman’s papers are cited in the manuscript. In addition, Reinhoudt’s article deals with metalo-dendrimeric, H-bonded rosettes, similar to those published by others (ref. [34-37] and the reviewing author ([38-39]) one decade later, duly described and reported in the manuscript. Regarding rosettes, even Whitesides is not mentioned.
2. Hydrogen-bonded arrays vs. metal coordination: Since H-bonds are weaker and less directional than metals, arrays of multiple hydrogen bonds are usually employed to shape the space and to control the strength of their primary and secondary interactions, forcing the ligands to adopt precise orientations/conformations for a successful assembly, A typical case is observed in calixarenes. For instance, giant cages have been described using uranyl and calixarene carboxylic acids (de Mendoza et al. Nat. Comm. 2012, 3, 785). Whereas calix[4]arenes can easily adopt the required cone conformation for octahedral capsule formation, no matter the lower rim is O-alkylated or unsubstituted, the related calix[5]arenes are only shaped conically if the lower rim is free, to produce a cyclic array of H-bonds that gives rise to a giant icosahedral uranyl cage from only 12 ligand molecules.
Also, quadruple H-bonds, as the author knows well, are required to shape the ligand dimers or aggregates into planar 2D structures for further build-up the metal-induced final structures. It is important to emphasize the control of the H-bonded structures, preventing unproductive rotations, in the designs.
The distinction of 2D from 3D structures in the titles of chapters 2 and 3 is misleading, as 2D is usually related to “flat”, which is almost always not the case in metal coordination spheres. In some other cases, the assembly is forced to rotate away from planarity due to steric constraints. For instance, structures of Fig 1 or Fig 2A could hardly be considered bidimensional. I suggest therefore use of the titles “macrocyclic…” and “macro(poly)cyclic and cages…” instead.
3. Compatibility and Solvent effects: It is well know that some metals brake H-bonding UPy dimers and related aggregates (work of Meijer and Sijbesma, also Reinhoudt…). This is part of the concept of orthogonality in supramolecular chemistry.
Also, polarity of solvents is critical as it can make some designs based on H-bonds and metal coordination not compatible. Strong H-bonded arrays (such as UPy dimers) resist substantial amounts of polar solvents (DMSO for instance) without breaking. In some exceptional cases, e.g. in various examples of ref. [50], the assemblies survive in methanol, or after addition of >50% water to DMSO solutions. These aspects should be mention, and better rationalized in the manuscript.
Author Response
I would like to thank the Reviewer for the positive comments, and the following changes have been made to improve the quality of the manuscript.
1) As requested by the Reviewer (also Reviewer 1), the orthogonality concept is now discussed, and the challenge to combine two different interactions better emphasized (introduction and conclusion). All suggested references, and a few more, were added to the revised manuscript.
2) The uranyl-based cages are now discussed, including a new figure (X-ray structure).
The titles of the sections have been replaced by "Planar and macrocyclic assemblies" and "Cage-like assemblies" as suggested.
3) When possible, compatibility and solvent effect were discussed. However, as the field remains relatively new, often, these aspects were not discussed by Authors. This is also why several systems are more described on "what" instead on "why", to make the manuscript easier to follow, and I'm sorry for that. However, the conclusion has been modified to point out these aspects, to ensure that the readers will better understand the challenges ahead.